# Adopting a model of antimicrobial stewardship program to anti-tubercular treatment stewardship: A single-centre experience from a private tertiary care hospital in South India

Swathy S. Samban[1☉], Akhilesh Kunoor[2☉], Preetha Prasanna[3], Malavika Krishnakumar[4], Nandita Shashindran[5], Chithira V. Nair[1], Abhinandh Babu[1], Ananth Ram K. J.[1], Sivapriya G. Nair[1], Subhash Chandra[6], Kiran G. Kulirankal[1], Georg Gutjahr[7], Rakesh P. S.[8], Dipu T. Sathyapalan[1]*, Merlin Moni[1]

1 Division of Infectious Diseases, Department of General Medicine, Amrita Institute of Medical Science and Research Centre, Amrita Vishwa Vidyapeetham, Kochi, Kerala, India, 2 Department of Respiratory Medicine, Amrita Institute of Medical Science and Research Centre, Amrita Vishwa Vidyapeetham, Kochi, Kerala, India, 3 Department of Medical Administration, Amrita Institute of Medical Science and Research Centre, Amrita Vishwa Vidyapeetham, Kochi, Kerala, India, 4 Department of Health Science Research, Amrita Institute of Medical Science and Research Centre, Amrita Vishwa Vidyapeetham, Kochi, Kerala, India, 5 Department of Microbiology, Amrita Institute of Medical Science and Research Centre, Amrita Vishwa Vidyapeetham, Kochi, Kerala, India, 6 Department of General Medicine, Amrita Institute of Medical Science and Research Centre, Amrita Vishwa Vidyapeetham, Kochi, Kerala, India, 7 Center for Research in Analytics & Technology for Education (CREATE), Amrita Vishwa Vidyapeetham, Amritapuri, Kollam, Kerala, India, 8 Senior Technical Advisor, The Union, South East Asia Office, New Delhi, India

☉ These authors contributed equally to this work.
* diputsmck@gmail.com

## Abstract

Tuberculosis (TB) remains a significant public health challenge in Low- and Middle-Income Countries (LMIC). Inappropriate use of Anti-Tubercular Treatment (ATT) undermines treatment efficacy and could contribute to drug resistance. While antimicrobial stewardship programs (AMSP) are well established, anti-tubercular treatment stewardship programs (ATTSP) in private hospitals do not have an established model. An AMSP model in a private tertiary care hospital in South India was repurposed to monitor the prescription appropriateness of ATT. A multidisciplinary team evaluated the ATT prescription appropriateness among inpatients over a one-year period with the 4R's criteria: Right Indication, Right Drug, Right Dose, and Right Frequency. The ATTSP team filed recommendations for inappropriate prescriptions to the primary clinical care team, and compliance to the recommendations was documented within 48 hours. During the study period, 172 ATT prescriptions were evaluated. Inappropriate dose and drug prescriptions were found in 16% and 7%, respectively. The primary clinical care teams complied with 83% of the recommendations within 48 hours. The potential impact of implementing the ATTSP model nationwide was assessed using published data, suggesting that the opportunities to correct inappropriate prescriptions could reach a quarter million. The study provides a proof of concept that an ATTSP can be

**Data Availability Statement:** All relevant data are within the manuscript and its Supporting information files.

**Funding:** The author(s) received no specific funding for this work.

**Competing interests:** The authors have declared that no competing interests exist.

successfully implemented in a TB endemic, resource-constrained setting. Extrapolation for implementing ATTSP across the country has the potential for huge public health benefits.

## Introduction

Tuberculosis (TB) continues to be a major global health issue, with India experiencing a disproportionate share of new cases and deaths [1]. More than one-fourth of the world's new TB cases are estimated to be in India, and 32% of estimated TB deaths worldwide occur in India [2, 3]. In the new National Strategic Plan (NSP), India has pledged to achieve the Sustainable Development Goals (SDG) related to ending TB by 2025, five years ahead of the global targets [4, 5].

The private health sector is massive but heterogeneous, and more than half the TB patients in India seek care from the private sector [6, 7]. Data on anti-TB medicine sales in India indicated that the majority of Anti-Tubercular Treatment (ATT) prescriptions are made from the private healthcare sector [8]. The suboptimal quality of TB care in the private sector is concerning, owing to non-standardised diagnosis and treatment protocols. Numerous efforts have been made to improve the quality of TB care in the private sector—System for TB Elimination in Private Sector (STEPS), being one such initiative. STEPS is a private-sector-led initiative to ensure uniform high standards of TB care for all the clients reaching them. It has streamlined TB-treatment notification, surveillance, contact tracing and TB Preventive Therapy, besides also ensuring patient access to social welfare services [9]. However, concerns remain regarding the quality of ATT prescriptions in the private sector [6]. Inappropriate ATT prescriptions worsen patient outcomes, in addition to implications for patient safety. Furthermore, they lead to the emergence of drug-resistant TB, [10] which further complicates TB care and efforts to end TB, especially in the context where India harbours the highest rates of multidrug-resistant TB (MDR-TB) in the world [11, 12].

Inappropriate prescriptions result from multiple causes, including considerable heterogeneity of ATT prescribers [13] with varying competence ranging from broad specialists, superspecialists and surgeons, frequent changes in TB-treatment guidelines, [14] increasing complexity of TB cases due to the associated comorbidities, [15, 16] an increasing burden of extrapulmonary TB, [17] paucity of continuous training of prescribers and the like [18].

For the related problem of antimicrobial prescriptions, a formal framework for measuring the appropriateness of antimicrobial prescriptions by 5R's is well established and validated. The 5R's encompass the Right indication, Right drug, Right dose, Right frequency and Right duration of the antimicrobials. Antimicrobial stewardship models have been accepted worldwide and have been shown to improve patient outcomes [19–21]. However, similar models do not exist for optimising ATT prescriptions.

While the awareness of Antimicrobial Resistance (AMR) is widespread and the mandatory need for having Antimicrobial Stewardship Programs (AMSP), [22] is well accepted, a similar issue in the setting of TB field is vastly neglected. One reason could be that while the majority of the patients monitored by the AMSP are inpatients, TB patients are mostly outpatients. Furthermore, the optimal ATT prescriptions are not static and patients have to be monitored throughout the treatment period to account for the weight-based dose modifications [23] and the adverse drug reactions [24].

We hypothesise that the existing antimicrobial stewardship models can be successfully repurposed for the Anti-Tubercular Treatment Stewardship Program (ATTSP) and their implementation would improve the appropriateness of the ATT prescriptions, lead to better

patient outcomes, reduce costs and the MDR-TB risk. This paper describes the feasibility and impact of implementing ATTSP over a one-year period in a tertiary care hospital in South India.

## Materials and methods

### Study design and population

This prospective observational study was conducted at the Amrita Institute of Medical Sciences and Research Centre, Kochi, Kerala, South India. This hospital is a 1300-bed tertiary care hospital with a structured clinical-pharmacist-driven antimicrobial stewardship program since 2016 [25]. Inpatients initiated on ATT drugs from November 2022 to December 2023 were included in the study. The process was implemented for ATT prescriptions of inpatients. Those treated on an outpatient basis and MDR-TB prescriptions were excluded.

### Patient consent statement

Ethics approval was duly obtained from the Institutional Ethics Committee (IEC-AIMS-2018-GENMED-097). Written informed consent was taken from all the study participants and for the 9 minor participants written consent was obtained from parents or guardians.

### Team and workflow of ATTSP

The ATTSP model (Fig 1) is repurposed from the existing AMSP protocol of the institution [25]. The AMSP methodology centres around a clinical-pharmacist-driven post-prescriptive audit of reserve antimicrobial prescriptions, in which daily reviews of patient data on restricted antibiotics are conducted to assess the appropriateness of therapy. This evaluation of

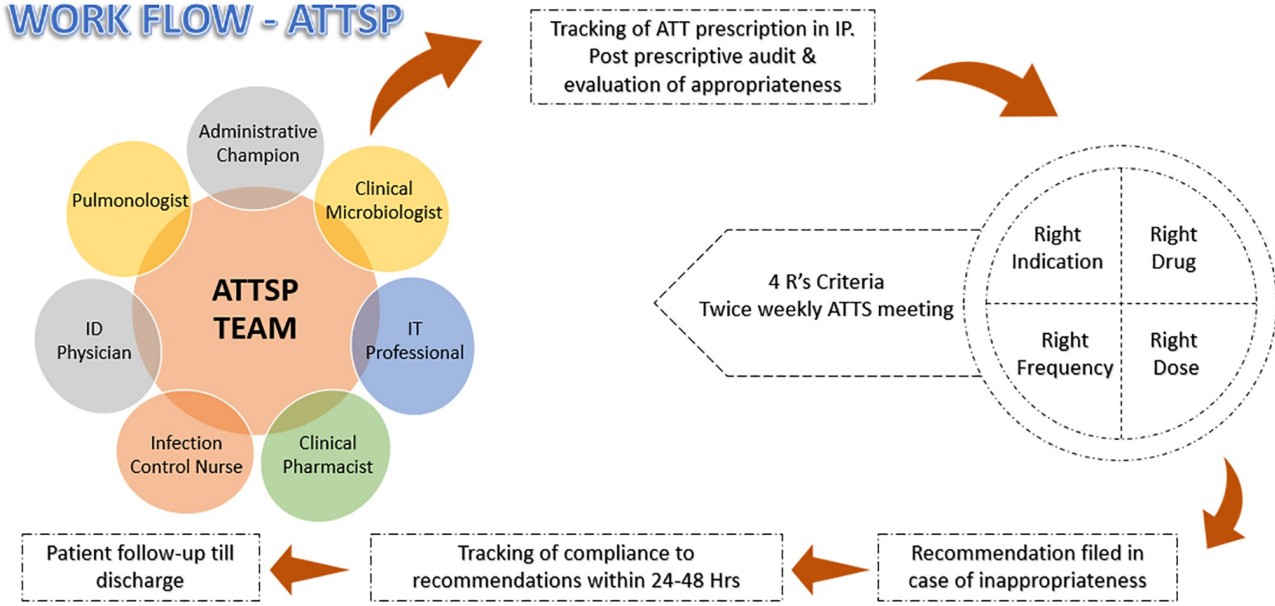

**Fig 1. ATT stewardship workflow.** ID Physician/Pulmonologist: Evaluation of appropriateness. Administrative champion: Putting together the team and facilitating the operational activities. Clinical Microbiologist: Identification and reporting of the TB cases. IT Professional: Running the report daily to identify reported TB cases and handing over the generated report to the ATTSP clinical pharmacist. Clinical Pharmacist: ATTSP tracking in Inpatients, post-prescriptive audit, tracking of compliance and patient follow-up till discharge. Infection Control Nurse (ICN): Ensuring appropriate isolation precautions.

appropriateness is grounded in the "5R's", adhering to guidelines from the Infectious Diseases Society of America (IDSA), The Society of Health Care Epidemiology of America (SHEA) and the United States Centres for Disease Control and Prevention (U.S. CDC). The multidisciplinary committee reviews prescriptions and suggests modifications based on the 5R's. Recommendations are recorded in the patient's file, and compliance with them is subsequently assessed to measure the program's effectiveness and the impact on antimicrobial use within the hospital [26].

A similar strategy of post-prescriptive audit and feedback was employed for the ATTSP. For this study, the "5R's" are tailored to "4R's" by exempting "Right duration", as the scope of the process did not encompass monitoring the entire treatment duration. The core team for ATTSP consisted of an Infectious Disease (ID) physician, a pulmonologist, an administrative champion, a clinical microbiologist, an Information Technology (IT) professional, a clinical pharmacist and an Infection Control Nurse(ICN) [27]. ATT prescriptions were identified from the pharmacy consumption records. The patient details were collected from the electronic medical records and bedside patient visits. Twice a week, the core team met to review the ATT prescriptions. Inappropriate prescriptions were notified to the primary clinical care team, and the recommendations from the ATTSP team were filed in a specially designed ATTSP recommendation form. The primary clinical care team was informed via telephone or in person, and the files were audited for compliance by the clinical pharmacist within the next 24–48 hours and up till the time of discharge.

## Study measurements

Demographic details, department at the time of admission, and comorbidities of the patients were collected from electronic case records and patient files. The TB cases were classified as pulmonary/extrapulmonary, based on the site of the infection. The TB was also classified into microbiologically confirmed cases, alongside histopathological and/or radiological suspected cases. New cases/recurrence/ relapse and reinfection were defined as per the technical and operational guidelines (TOG) guidelines of National Tuberculosis Elimination Programme (NTEP) [28]. The rightness of indication, drug, dose and frequency has been assessed based on the Standards of TB Care in India, NTEP guidelines and INDEX TB guidelines [29].

## Statistical methods

Baseline characteristics are summarised by counts and percentages and stratified by the type of TB. The TB patient distribution is presented for the different medical and surgical departments. For each department, the proportion of inappropriate prescriptions along with recommendations and compliance is reported. The numbers of inappropriate cases and recommendations are stratified according to the 4R's. The results of the microbiological tests are also expressed in proportions. To extrapolate the findings to the private health sector in India, TB incidence and patient number estimates were obtained from NTEP.

## Results

In total, 172 patients met the inclusion criteria of the study. The median age of the patients was 54 years ((IQR) Inter Quartile Range 40 to 65). The median length of hospital stay was 10 days (IQR 6 to 15). Extrapulmonary TB constituted 65.12%. Table 1 shows the distribution of the baseline characteristics. The most common comorbidities were diabetes, followed by hypertension and Coronary Artery Disease(CAD). Most TB cases were "new cases," with recurrent TB amounting to only 6.40%.

**Table 1. Baseline characteristics of the study patients stratified by the type of TB.**

| Characteristics | Overall | | Extra Pulmonary | | Pulmonary | |
|---|---|---|---|---|---|---|
| | n | % | n | % | n | % |
| Total | 172 | 100 | 112 | 65.12 | 60 | 34.88 |
| Age > 60 | 68 | 39.53 | 47 | 69.12 | 21 | 30.88 |
| Female | 61 | 35.47 | 51 | 83.61 | 10 | 16.39 |
| Weight > 60kg | 89 | 51.74 | 53 | 59.55 | 36 | 40.45 |
| Length of stay > 7D[a] | 113 | 65.7 | 84 | 74.34 | 29 | 25.66 |
| Diabetes | 66 | 38.37 | 38 | 57.58 | 28 | 42.42 |
| Hypertension | 57 | 33.14 | 37 | 64.91 | 20 | 35.09 |
| CAD[b] | 20 | 11.63 | 10 | 50 | 10 | 50 |
| CKD[c] | 17 | 9.88 | 10 | 58.82 | 7 | 41.18 |
| CLD[d] | 4 | 2.33 | 2 | 50 | 2 | 50 |
| Dyslipidemia | 14 | 8.14 | 9 | 64.29 | 5 | 35.71 |
| Hypothyroidism | 8 | 4.65 | 7 | 87.5 | 1 | 12.5 |
| Malignancy | 5 | 2.91 | 3 | 60 | 2 | 40 |
| Asthma | 4 | 2.33 | 2 | 50 | 2 | 50 |
| Other comorbidities[e] | 15 | 8.72 | 9 | 60 | 6 | 40 |
| History of COVID-19[f] | 22 | 12.79 | 13 | 59.09 | 9 | 40.91 |
| Recurrent TB cases | 11 | 6.4 | 6 | 54.55 | 5 | 45.45 |

[a]Days

[b]Coronary Artery Disease

[c]Chronic Kidney Disease

[d]Chronic Liver Disease

[e]Chronic Obstructive Pulmonary Disease (COPD), Cerebral Vascular Accident (CVA), seizure disorder, Human Immunodeficiency Virus (HIV), Systemic Lupus Erythematosus (SLE), and rheumatoid arthritis

[f]CoronaVirus Disease of 2019

Table 2 shows the primary admission departments of TB patients. Of the total 172 patients, 42 had inappropriate prescriptions. Among the 42 cases with inappropriate prescriptions, recommendations were filed for 36, whereas for six, no recommendations were filed as three patients were discharged early and three expired in-hospital. As filing of recommendations for the patients who were discharged early or who expired were not feasible, feed back to the clinicians were given by the team over the phone or in person. Compliance with the recommendation was 83.33% checked within 48 hours.

Table 3 shows the proportion of inappropriateness in accordance with the 4 R's. The most common recommendation was optimising the ATT dose. Among the 27 recommendations filed for dose adjustment, 15 were for inappropriately high doses and 12 for inappropriately low doses.

17 patients developed Adverse Drug Reaction (ADR) during in hospital stay and, the most observed ADR was ATT induced hepatitis (threefold increase in Serum Glutamic Oxaloacetic Transaminase (SGOT) and Serum Glutamate Pyruvate Transaminase (SGPT) values) S1 Table.

It was observed that an increase in age was associated with a higher likelihood of dosing errors S2 Table.

Table 4 shows the three microbiological tests, namely GeneXpert, acid-fast bacilli culture, and AFB smear and the positivity rate of individual tests and their combinations.

**Table 2. Department-wise distribution of patients in each department along with inappropriate prescriptions and the recommendations filed.**

| Department | Overall | | Inappropriate | | Recommendations | |
|---|---|---|---|---|---|---|
| | n | % | n | % | n | % |
| General Medicine | 43 | 25 | 12 | 27.91 | 12 | 100 |
| Respiratory Medicine | 26 | 15.12 | 6 | 23.08 | 5 | 83.33 |
| Surgery | 25 | 14.53 | 5 | 20 | 4 | 80 |
| Pulmonary Medicine | 20 | 11.63 | 3 | 15 | 3 | 100 |
| Neurology | 10 | 5.81 | 5 | 50 | 4 | 80 |
| Cardiology | 9 | 5.23 | 3 | 33.33 | 2 | 66.67 |
| Nephrology | 9 | 5.23 | 3 | 33.33 | 2 | 66.67 |
| Gastroenterology | 7 | 4.07 | 2 | 28.57 | 2 | 100 |
| Orthopedics | 6 | 3.49 | 2 | 33.33 | 1 | 50 |
| Others[a] | 17 | 9.88 | 1 | 5.88 | 1 | 100 |

[a]Urology, Haematology, Rheumatology, Psychiatry, Cardio Vascular and Thoracic Surgery (CVTS), and Medical Oncology

## Discussion

Our prospective observational study shows how an AMSP model can be effectively repurposed to monitor the prescription appropriateness of ATT. It provides a proof of concept that an ATTSP can be successfully implemented in a TB endemic, resource-constrained Low- and Middle-Income Country (LMIC) setting. It also demonstrates how the plan for a prescription audit envisioned in the NSP for ending TB in India could be materialised. While AMSP has gained traction globally as a means to improve patient outcomes and curb resistance, [20] no similar programs existed for ATT prescriptions; to our knowledge, this is the first work that proposes a formal model for ATTSP.

**Table 3. Distribution of 4 R's (Right Dose, Right Drug, Right Frequency, and Right Indication) with the frequency of inappropriateness and recommendations filed for each R.**

| Characteristics | Inappropriate | | Recommendations | | Compliances | |
|---|---|---|---|---|---|---|
| | n | % | n | % | n | % |
| Dose | 27* | 15.70 | 23 | 85.19 | 19 | 82.61 |
| Drug | 12 | 6.98 | 11 | 91.67 | 7 | 63.64 |
| Frequency | 3 | 1.74 | 3 | 100.00 | 3 | 100.00 |
| Indication | 3 | 1.74 | 2 | 66.67 | 1 | 50.00 |

**Table 4. Microbiological test for diagnosis of TB for each test and combination of tests: The total number along with the positive and negative cases are shown.**

| Microbiological tests | Overall | | | | Pulmonary TB | | | | Extra Pulmonary TB | | | |
|---|---|---|---|---|---|---|---|---|---|---|---|---|
| | Tested | | Positive | | Tested | | Positive | | Tested | | Positive | |
| | n | % | n | % | n | % | n | % | n | % | n | % |
| GeneXpert | 160 | 93.02 | 100 | 62.50 | 58 | 96.67 | 50 | 86.21 | 102 | 91.07 | 50 | 49.02 |
| Culture | 118 | 68.60 | 41 | 34.75 | 38 | 63.33 | 18 | 47.37 | 80 | 71.43 | 23 | 28.75 |
| Smear | 145 | 84.30 | 26 | 17.93 | 54 | 90.00 | 20 | 37.04 | 91 | 81.25 | 6 | 6.59 |
| GeneXpert or Culture or Smear | 165 | 95.93 | 111 | 67.27 | 59 | 98.33 | 53 | 89.83 | 106 | 94.64 | 58 | 54.72 |

By focusing on the "4R's"—Right Indication, Right Drug, Right Dose and Right Frequency —through the ATTSP model, the study highlights the potential for identifying inappropriate prescriptions. Although the primary strategy involves prospective audit of the prescriptions, feedback to the primary prescribers is ensured through filing of recommendation form or direct feedback via in person or over the phone. The compliance to the recommendations will be followed up within 24 to 48-hour time period. In the public sector, the standardisation of ATT prescriptions is facilitated by programs such as NTEP and the use of Fixed Dose Combinations (FDCs). However, in the private sector, prescriptions are more personalised due to comorbidities and adverse drug reactions, complicating standardisation. This complexity is amplified in the current study by the high number of extrapulmonary TB cases, as it functions as a tertiary care referral centre. Among the audited ATT prescriptions, dug induced hepatitis constituted the most common ADR during in hospital stay and adjustments to the dose and drug selection were the most frequent recommendations filed. The majority of dosing errors were attributed to incorrect dose calculations. Specific issues included doses exceeding the maximum recommended dose and instances of under dosing even in scenarios where patients show mildly deranged liver enzymes. Additionally, there were cases where frequency of administration of drug were not considered in patients with chronic kidney disease (CKD) S3 Table.

In India, the private sector manages half of the TB patients, [6] and extrapolation of the study findings to the NTEP data shows that scaling ATTSP across the private sector can potentially rectify a quarter million inappropriate prescriptions by 2025. The NTEP goal of ending TB requires the continuous engagement of the private sector, and the ATTSP could play a crucial role. The ATTSP potential is maximised when complemented by systemic initiatives like STEPS, which tackles the fragmentation of TB care and the issue of lost-to-follow-up patients.

To further reinforce the impact of ATTSP and STEPS initiatives, it is imperative to focus on enhancing care quality through strategic surveillance and feedback mechanisms. While efforts to bridge this gap through public-private partnerships, challenges persist in ensuring the delivery of quality TB care across the healthcare sectors. Hospital-crediting organisations could mandate private hospitals to implement quality initiatives like ATTSP for standardised TB management. This step could pave the way for excellence in TB care across the nation.

While this model could be easily scalable (Fig 2) to all super speciality private hospitals in the country, which may be around 6000 in number, it needs further customisation for smaller hospitals and individual practitioners.

A centralized IT-based platform for providing feedback could greatly benefit smaller hospitals and individual practitioners. An electronic consilium could enhance the management of TB by offering scientifically sound, evidence-based clinical advice through the internet. This approach, modelled after the ERS/WHO E-consilium initiative for drug resistant TB, could be adopted at the national level to optimize ATT in high TB burden countries like India [30].

This study lays the groundwork for future research aimed at optimising TB treatment and preventing the emergence of MDR-TB strains. Research exploring the scalability of ATTSP models across different healthcare contexts will provide valuable insights for TB control strategies within India and also in other LMIC settings with a high-TB burden.

Fig 2 shows the extrapolation for implementing ATTSP across the country. According to the NSP, the number of TB patients across the country will reach 2 million by 2025, with half of them being notified from the private health sector. On extrapolating the observed opportunities from this single-centre study to the potential number of patients who would seek TB treatment in private hospitals across the nation by the year 2025, the opportunities to correct inappropriate prescriptions would reach a minimum of a quarter million.

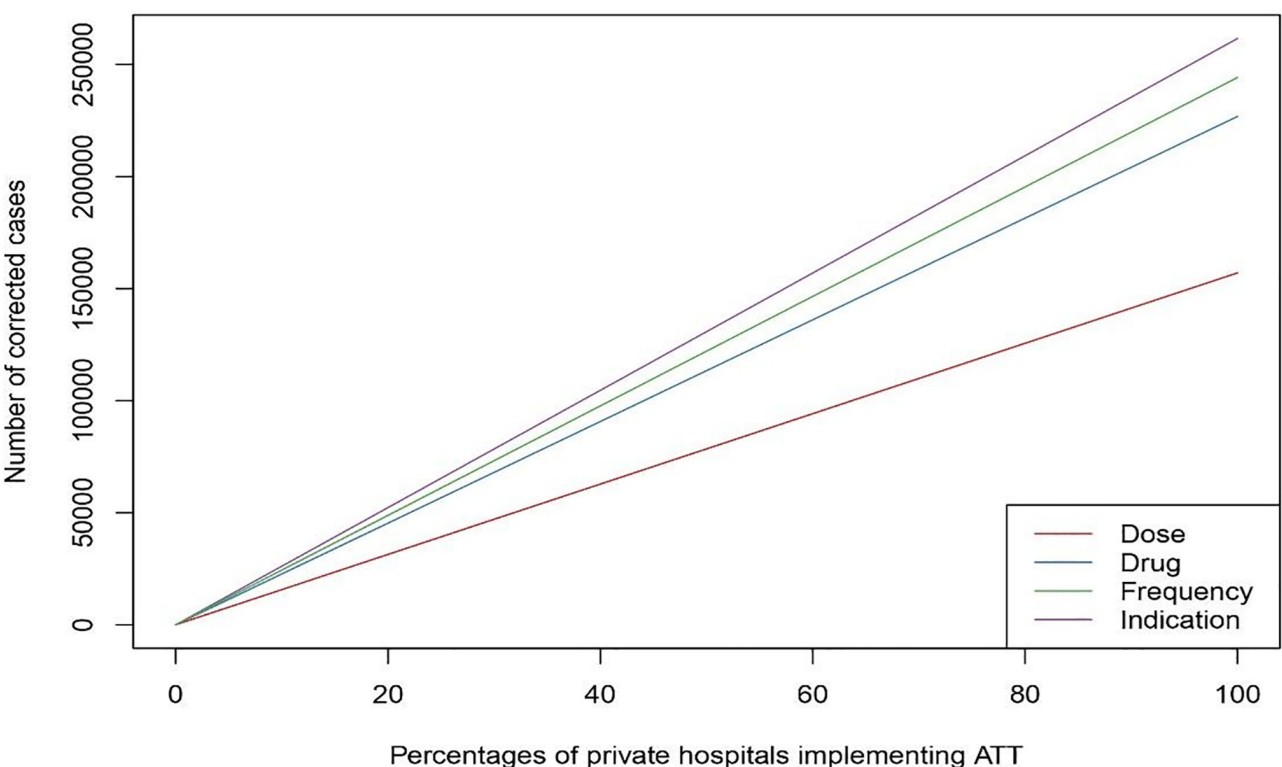

**Fig 2. Extrapolating the benefits of implementing ATT across the private hospitals in India.**

## Limitations

Our study was a single-centre study from a tertiary care centre, and hence extrapulmonary cases constituted the majority of the cohort. Although the appropriateness of ATT duration was not included in the current model, Right duration can be included in future models if there is a robust system to follow up patients for extended periods. Similarly, expanding the model to include outpatients and Drug-resistant TB (DR-TB) would improve the rollout to bigger populations as well.

## Conclusion

This study provides a proof of concept of the feasibility of repurposing the AMSP model for ATTSP. Scaling up this approach at a national level can potentially rectify a quarter million inappropriate prescriptions by 2025. Integrating ATTSP with current initiatives like STEPS in India's private sector represents a forward-thinking approach to eliminate TB.

## Supporting information

**S1 File. Data collection form.**
(PDF)

**S2 File. Recommendation form.**
(PDF)

**S1 Table. Adverse Drug Reactions (ADRs).**
(DOCX)

**S2 Table. Age-dose relation.**
(DOCX)

**S3 Table. Recommendations filed for 36 cases.**
(DOCX)

## Acknowledgments

We would like to express our sincere gratitude to all those who contributed to the successful completion of this study. We extend our heartfelt thanks to the Medical Administration for the valuable guidance and continuous support throughout this study which is a Quality improvement Initiative. Special thanks to the practitioners, clinical pharmacists, nursing staffs and all healthcare professionals at Amrita Institute of Medical Science and Research Centre, Amrita Vishwa Vidyapeetham, Kochi, Kerala, India for their collaboration and willingness to comply with the recommendation recognizing the value to the patient care, which played a crucial role in the implementation of the stewardship program. Lastly, we would like to acknowledge the patients and their families for their participation and cooperation in this study. Their willingness to contribute to this research is greatly appreciated.

## Author Contributions

**Conceptualization:** Rakesh P. S., Dipu T. Sathyapalan, Merlin Moni.

**Data curation:** Swathy S. Samban, Akhilesh Kunoor, Preetha Prasanna, Chithira V. Nair, Abhinandh Babu, Ananth Ram K. J., Sivapriya G. Nair.

**Formal analysis:** Swathy S. Samban, Akhilesh Kunoor, Malavika Krishnakumar, Georg Gutjahr.

**Methodology:** Subhash Chandra, Kiran G. Kulirankal, Georg Gutjahr, Rakesh P. S., Merlin Moni.

**Project administration:** Preetha Prasanna.

**Supervision:** Swathy S. Samban, Akhilesh Kunoor, Preetha Prasanna, Nandita Shashindran, Subhash Chandra, Kiran G. Kulirankal, Rakesh P. S., Dipu T. Sathyapalan, Merlin Moni.

**Validation:** Subhash Chandra, Kiran G. Kulirankal, Rakesh P. S., Dipu T. Sathyapalan, Merlin Moni.

**Visualization:** Rakesh P. S., Dipu T. Sathyapalan, Merlin Moni.

**Writing – original draft:** Swathy S. Samban, Akhilesh Kunoor, Preetha Prasanna, Georg Gutjahr.

**Writing – review & editing:** Swathy S. Samban, Akhilesh Kunoor, Malavika Krishnakumar, Chithira V. Nair, Abhinandh Babu, Ananth Ram K. J., Sivapriya G. Nair, Subhash Chandra, Kiran G. Kulirankal, Georg Gutjahr, Rakesh P. S., Dipu T. Sathyapalan, Merlin Moni.

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
