## [Decision Letter · Decision Letter 0]

16 Jul 2024

PONE-D-24-23210Adopting a Model of Antimicrobial Stewardship Program to Anti-Tubercular Therapy Stewardship: A Single-CentreExperience from a Private Tertiary Care Hospital in South IndiaPLOS ONE

Dear Dr. T Sathyapalan,

Thank you for submitting your manuscript to PLOS ONE. After careful consideration, we feel that it has merit but does not fully meet PLOS ONE’s publication criteria as it currently stands. Therefore, we invite you to submit a revised version of the manuscript that addresses the points raised during the review process.

We look forward to receiving your revised manuscript.

Kind regards,

Arghya Das, MD

Academic Editor

PLOS ONE

Journal Requirements:

2. Please provide additional details regarding participant consent. In the ethics statement in the Methods and online submission information, please ensure that you have specified (a) whether consent was informed and (b) what type you obtained (for instance, written or verbal, and if verbal, how it was documented and witnessed). If your study included minors, state whether you obtained consent from parents or guardians. If the need for consent was waived by the ethics committee, please include this information.

**Additional Editor Comments:**

Authors of the present manuscript presents a proof of concept for antimicrobial stewardship in tuberculosis which they have mentioned as Anti-Tubercular Therapy Stewardship (ATTSP). From the same institute, a previously published article mentioned the term "anti-tuberculosis treatment stewardship". As a new concept is being presented, it is advisable if the authors could use uniform terminologies to describe the concept.

Although the concept presented in the manuscript is new and pertinent in contemporary landscape of TB treatment (with special emphasis to private sector hospitals), it appears to be more of a prescription audit. The exclusion of one of the 5Rs 'Right Duration' from the assessment is justifiable due to lack of follow up data after discharge, consideration of few important aspects like monitoring adverse drug reactions during inpatient treatment could make the study more meaningful.

Please address the above in the revised version of the manuscript under DISCUSSION.

MATERIALS AND METHODS

Line 101: This prospective cohort study was conducted.................

Comment: The study design does not seem to be that of a prospective cohort study. Please mention it as a descriptive observational study only.

Line 114: ...........Centres for Disease Control and Prevention (CDC)

Comment: Please mention the above as United States CDC

Line 143: Fig 2: Extrapolating the benefits of implementing ATT across the private hospitals in India

Comment: Figure 2 and its legends have been mentioned twice in the manuscript; once under MATERIALS AND METHODS and second time under DISCUSSION

Line 156-158: New cases/recurrence/ relapse..................................National Tuberculosis Elimination Programme (NTEP)

Comment: Please cite the suitable reference for the above sentence.

RESULTS

Lines 184-185: ..................no recommendations were filed as three patients were discharged early.

Comment: It is not understood why recommendations could not be filed for three patients who were discharged. In the event of detection of inappropriate prescription, it's a sine qua non to rectify the same to avoid further complications. Please explain.

Lines 193-196: The most common recommendation was optimising the ATT dose. Among the 27 recommendations filed for dose adjustment, 15 were for inappropriately high doses and 12 for inappropriately low doses.

Comment: The above statistics require more elaboration. Was there any effect of age-group (like children) on the high doses or low doses? That means, whether the wrong prescribed doses may be resulting from not-considering dose-adjustments or wrongly calculating the doses. If such instances are documented please include them in the revised manuscript.

In the same context, it is also advisable to add the demographic details, drugs prescribed including the inappropriate ones and inappropriate doses prescribed, inappropriate frequency and inappropriate indication in a tabular format for those 36 cases where recommendations were filed. Please upload this information as a Supplementary file.

Lines 237-239: ...................so an information-technology-enabled platform for providing................................associations or city consortiums of private providers may be developed.

Comment: One such IT-based platform ERS/WHO TB Consilium to provide rapid advice to clinicians already exists. Please consider adding it under the same paragraph adding suitable references.

TABLE 1: Tables should be self explanatory. Please expand all abbreviations in the footnote.

FIGURES: All figures should also be cited sequentially in the text.

Figure 2 seems to be reproduced from other sources. Please acknowledge the source and cite the reference accordingly.

Reviewers' comments:

Reviewer's Responses to Questions

**Comments to the Author**

1. Is the manuscript technically sound, and do the data support the conclusions?

Reviewer #1: Yes

Reviewer #2: Yes

2. Has the statistical analysis been performed appropriately and rigorously? 

Reviewer #1: Yes

Reviewer #2: Yes

3. Have the authors made all data underlying the findings in their manuscript fully available?

Reviewer #1: Yes

Reviewer #2: Yes

4. Is the manuscript presented in an intelligible fashion and written in standard English?

Reviewer #1: Yes

Reviewer #2: Yes

5. Review Comments to the Author

**Reviewer #1:** As TB is increasing burden in our country , we have to strengthen the treatment by implementing ATTSP. Drug therapy for TB take longer duration, so we need to titrate the treatment course well organised without have any misdirections. Proper drug therapy will enhance the treatment outcome.

**Reviewer #2:** its a novel concept especially for developing countries where rampant use of ATT drugs has happened leading to surge in MDR and XDR cases of TB. The article is well written and easy to understand. The workflow can be implemented at other places too.

6. PLOS authors have the option to publish the peer review history of their article (what does this mean?). If published, this will include your full peer review and any attached files.

Reviewer #1: **Yes: **Dr M R Vasanthapriyan

Reviewer #2: **Yes: **Dr VIKRAMJEET SINGH, MD, DNB Microbiology, Assistant Professor, SGPGI Lucknow

---

## [Author Response · Author response to Decision Letter 0]

29 Aug 2024

Dear Reviewer’s,

We sincerely thank you for the time and effort you have invested in reviewing our manuscript. We are grateful for your constructive feedback, which has helped us enhance the quality and clarity of our work.

We have carefully addressed each of the points raised in your review and have made the necessary revisions to the manuscript. We have provided detailed responses to each comment, and added supplementary materials where appropriate.

Additionally, we have updated the author list to include Dr. Binny P. Prabhu and Dr. Nandita Shashindran acknowledging their significant contributions to the study.

We trust that these revisions meet your expectations and contribute to the overall improvement of our manuscript. We look forward to your further insights.

Thank you once again for your valuable feedback.

Journal Requirements:

Carefully reviewed and formatted the manuscript according to the PLOS ONE's style requirements

2. Please provide additional details regarding participant consent. In the ethics statement in the Methods and online submission information, please ensure that you have specified (a) whether consent was informed and (b) what type you obtained (for instance, written or verbal, and if verbal, how it was documented and witnessed). If your study included minors, state whether you obtained consent from parents or guardians. If the need for consent was waived by the ethics committee, please include this information.

ANS: Thank you for your comments. In our study, written informed consent was obtained from all participants. For the nine paediatric participants included in the study, written consent was duly obtained from their parents or legal guardians. This has been clarified in the Methods section under the heading Patient Consent Statement. We will also ensure that this information is appropriately reflected in the online submission details. Line numbering: 117 - 120

This study was not a retrospective study. Informed written consent was obtained from all participants prospectively, prior to data collection. 

ANS: Thank you for the careful review of the manuscript and insightful comments. The reference list as requested was reviewed and confirmed that none of the cited papers have been retracted. To enhance the manuscript, additional reference has been added that is highly relevant to the topic discussed: 

Singh S, Menon VP, Mohamed ZU, Kumar VA, Nampoothiri V, Sudhir S, Moni M, Dipu TS, Dutt A, Edathadathil F, Keerthivasan G. Implementation and impact of an antimicrobial stewardship program at a tertiary care center in South India. In Open Forum Infectious Diseases 2019 Apr (Vol. 6, No. 4, p. ofy290). US: Oxford University Press.

This reference is particularly important as it details antimicrobial stewardship model implemented at the same institution, providing a comprehensive context and supporting the findings presented in the manuscript. All changes to the reference list have been reflected in the revised manuscript, as reference number 25.

Additional Editor Comments:

1. A) Authors of the present manuscript presents a proof of concept for antimicrobial stewardship in tuberculosis which they have mentioned as Anti-Tubercular Therapy Stewardship (ATTSP). From the same institute, a previously published article mentioned the term "anti-tuberculosis treatment stewardship". As a new concept is being presented, it is advisable if the authors could use uniform terminologies to describe the concept.

ANS 1.A) Thank you for your valuable feedback. We understand the importance of using uniform terminologies to describe the concept. In light of your suggestion, we have revised the terminology in our manuscript to "anti-tubercular treatment stewardship program (ATTSP)" to ensure consistency with previous publications from our institute, along with the fact that ATT is expanded as “Anti Tubercular Treatment” in RNTCP guidelines. We believe this will help maintain clarity and uniformity in the literature.

B) Although the concept presented in the manuscript is new and pertinent in contemporary landscape of TB treatment (with special emphasis to private sector hospitals), it appears to be more of a prescription audit. 

ANS 1.B) Thank you for your feedback. While we understand that the concept may appear similar to a prescription audit, it is more accurately described as a stewardship model that was adopted. Specifically, it represents the widely accepted stewardship strategy of prospective audit and feedback. This approach includes a prospective audit and feedback mechanism, along with tracking compliance, which distinguishes it from a simple prescription audit. To ensure clarity, we have now added the explanation of this strategy to the discussion section of the manuscript. Line numbering 253-256.

C)The exclusion of one of the 5Rs 'Right Duration' from the assessment is justifiable due to lack of follow up data after discharge, 

ANS 1.C) The exclusion of the 'Right Duration' from our assessment is indeed due to the lack of follow-up data post-discharge, as mentioned. We have already acknowledged this in the limitations section under the discussion section of the manuscript. Line numbering.310-311

D) Consideration of few important aspects like monitoring adverse drug reactions during inpatient treatment could make the study more meaningful.

Please address the above in the revised version of the manuscript under DISCUSSION.

ANS 1.D) Thank you for your insightful suggestion regarding the inclusion of adverse drug reaction (ADR) monitoring. We have now addressed this in the revised manuscript. This information has been incorporated into both the results and discussion sections and has also been added as supporting information (S1 Table). Line numbering 228-230, 261-262

2. MATERIALS AND METHODS

Line 101: This prospective cohort study was conducted.................

Comment: The study design does not seem to be that of a prospective cohort study. Please mention it as a descriptive observational study only.

ANS 2: Thank you for the comment. We agree that the study design is more accurately described as an observational study. To reflect this, we have revised the manuscript to describe the study as a “prospective observational study”. Line numbering: 110,243

3. Line 114: ...........Centres for Disease Control and Prevention (CDC)

Comment: Please mention the above as United States CDC

ANS 3: Thank you for your suggestion. We have revised the manuscript to specify "United States CDC" as requested. Line numbering: 129

4. Line 143: Fig 2: Extrapolating the benefits of implementing ATT across the private hospitals in India

Comment: Figure 2 and its legends have been mentioned twice in the manuscript; once under MATERIALS AND METHODS and second time under DISCUSSION

ANS: Thank you for bringing this to our attention. We will revise the manuscript to ensure Figure 2 and its legends are mentioned only once, under the DISCUSSION section. 

5. Line 156-158: New cases/recurrence/ relapse..................................National Tuberculosis Elimination Programme (NTEP)

Comment: Please cite the suitable reference for the above sentence.

ANS: Thank you for pointing out the need for citation. Appropriate reference added to support the statement regarding new cases, recurrence, and relapse in the context of the National Tuberculosis Elimination Programme (NTEP). Ref number 28.

6. RESULTS

Lines 184-185: ..................no recommendations were filed as three patients were discharged early.

Comment: It is not understood why recommendations could not be filed for three patients who were discharged. In the event of detection of inappropriate prescription, it's a sine qua non to rectify the same to avoid further complications. Please explain.

ANS: Thank you for your comment. The filing of recommendations refers to documenting them directly in the patients' files, a process primarily conducted for inpatients. For the three patients who were discharged early, direct filing was not feasible. However, in these instances, the clinicians were promptly informed of the recommendations either in person or over the phone. This clarification has been addressed in the results section of the manuscript. Line numbering: 211-213

7. Lines 193-196: The most common recommendation was optimising the ATT dose. Among the 27 recommendations filed for dose adjustment, 15 were for inappropriately high doses and 12 for inappropriately low doses.

Comment: The above statistics require more elaboration. Was there any effect of age-group (like children) on the high doses or low doses? That means, whether the wrong prescribed doses may be resulting from not-considering dose-adjustments or wrongly calculating the doses. If such instances are documented please include them in the revised manuscript.

In the same context, it is also advisable to add the demographic details, drugs prescribed including the inappropriate ones and inappropriate doses prescribed, inappropriate frequency and inappropriate indication in a tabular format for those 36 cases where recommendations were filed. Please upload this information as a Supplementary file.

ANS: Effect of Age-Group on Dose Errors: In our analysis we observed that an increase in age was associated with a higher likelihood of dosing errors. This analysis has been detailed in the supporting information (S2 Table). Line numbering: 231-232

 Causes of Incorrect Dosing: The majority of dosing errors were attributed to incorrect dose calculations. Specific issues included doses exceeding the maximum recommended dose and instances of underdosing even in scenarios where patients show mildly deranged liver enzymes. Additionally, there were cases where frequency of administration of drug were not considered in patients with chronic kidney disease (CKD). This is now included in the discussion section. Line numbering. Line numbering:263-267

We have added a supplementary table providing detailed demographic information, including the drugs prescribed (both appropriate and inappropriate), inappropriate doses, frequencies, and indications for the 36 cases where recommendations were filed. This table is now included as supplementary material to offer a comprehensive view of the dosing issues encountered. Added as supporting information (S3 Table).

8. Lines 237-239: ...................so an information-technology-enabled platform for providing................................associations or city consortiums of private providers may be developed.

Comment: One such IT-based platform ERS/WHO TB Consilium to provide rapid advice to clinicians already exists. Please consider adding it under the same paragraph adding suitable references.

ANS: Thank you for your valuable suggestion. We have incorporated it to the manuscript with suitable reference. Line numbering: 283-287. Reference number:30

9. TABLE 1: Tables should be self explanatory. Please expand all abbreviations in the footnote.

Thank you for the suggestion. The abbreviations in Table 1 have been expanded in the footnote to ensure the table is self-explanatory.

10. FIGURES: All figures should also be cited sequentially in the text.

Figure 2 seems to be reproduced from other sources. Please acknowledge the source and cite the reference accordingly.

ANS: Thank you for your feedback. The figures have been cited sequentially in the text. For Figure 2, the statistics used for extrapolation were taken from the National Strategic Plan, mentioned accordingly in the manuscript and the figure itself is an original creation.

---

## [Editor Report · Decision Letter 1]

2 Sep 2024

Adopting a model of antimicrobial stewardship program to anti-tubercular treatment stewardship: A single-centreexperience from a private tertiary care hospital in South India

PONE-D-24-23210R1

Dear Dr. T Sathyapalan,

We’re pleased to inform you that your manuscript has been judged scientifically suitable for publication and will be formally accepted for publication once it meets all outstanding technical requirements.

Kind regards,

Arghya Das, MD

Academic Editor

PLOS ONE
---

## [Editor Report · Acceptance letter]

16 Oct 2024

PONE-D-24-23210R1 

PLOS ONE

Dear Dr. Sathyapalan, 

I'm pleased to inform you that your manuscript has been deemed suitable for publication in PLOS ONE. Congratulations! Your manuscript is now being handed over to our production team.

Kind regards, 

on behalf of

Dr. Arghya Das 

Academic Editor

PLOS ONE